# Improving A Large Healthcare System Research Data Warehouse Using OHDSI's Data Quality Dashboard

Benoit L. Marteau[†], Andrew Hornback[†], Yishan Zhong[†], Christian Lowson[§], Jason Woloff[§],
Benjamin M. Smith[§], Coleman Hilton[§], May D. Wang[†]

[†] Georgia Institute of Technology, Atlanta, GA 30332
Email: benoitmarteau@gatech.edu; ahornback6@gatech.edu; yzhong307@gatech.edu; maywang@gatech.edu
[§] Shriners Hospitals for Children, Tampa, FL, USA
Email: christian.lowson@shrinenet.org; jason.woloff@shrinenet.org; benjaminm.smith@shrinenet.org; chilton@shrinenet.org

*Abstract*—In this study, we have focused on AI Implementation Science research to explore how multimodal medical data from multiple hospitals can be harmonized in the real world clinical setting to improve patient care. IN implementing AI for healthcare, a major challenge is the lack of clear identification of AI Implementation Science methods that are systematic from the ones that are case-specific minor fixes. We used the state-of-the-art HL7 Standard Fast Health Interoperability Resource (FHIR) to upgrade the SC clinical research informatics infrastructure; we conducted a comprehensive study of the SC Research Data Warehouse (RDW) using the Observational Health Data Sciences and Informatics (OHDSI) Data Quality Dashboard (DQD); we reported the completeness and conformity of the data to the Observational Medical Outcomes Partnership (OMOP) Common Data Model (CDM), and the data content plausibility; and we identified the failure. We also built a visual dashboard as a detailed feedback mechanism to help Shriners' researchers evaluate data quality. With these major implementation science issues discovered and solved, we not only improved the access and quality of SC RDW data to enable clinicians and researchers to more trustworthy decision support for better patient care outcome, but more importantly, we have developed an entire workflow and pipeline that can be extended and utilized for other healthcare systems.

*Index Terms*—OMOP CDM, OHDSI, health informatics, data harmonization, healthcare infrastructure, Data Quality Dashboard, database quality

## I. INTRODUCTION

### A. Background and Motivation

In current health systems around the world, huge volume of patient care data collected in different hospitals have largely been sitting in silo without common data standard for archival and retrieval. As the result, it is very hard for care providers to get holistic view of the patient health progression over time and over different locations In addition, patient care data are typically in multi-modality, including radiology and pathology imaging, photographic imaging, genomics, time series data, continuous physiological monitoring, motion tracking, demographics, and lab data etc. in Electronic Health Record (EHR). How to extract value out of such complex multi-modality data requires clinical decision support tools based on AI. Thus, part of the emerging and fast-growing AI Implementation Science research is to investigate how to establish secure and safe data harmonization environment so that AI-Driven clinical decision support systems developed will be trustworthy.

In this work, we have selected Shriners Children's (SC) as the platform for conducting AI Implementation Science research. SC is an international pediatric health system with 22 hospitals across US, Canada, and Mexico. It is well known for pediatric care in orthopedics, craniofacial disorders, and burn injuries, and does generate huge volume of multi-modality data. However, access to raw data in 22 hospitals sitting in three countries is hard with multiple challenges: how to maintain secure data within a closed system, how to have system-wide data harmonization and standardization, how to access data as care providers who do not know programming languages (e.g. Structured Query Language, SQL), how to enable clinicians to identify patient cohorts across multiple SC hospitals to assist in clinical decision making, and how to make the collaboration easier. SC has used Observational Health Data Sciences and Informatics' (OHDSI) Observational Medical Outcomes Partnership (OMOP) Common Data Model (CDM) as the Research Data Warehouse (RDW) standard. SC data engineers have established an Extract Load Transfer (ETL) process to migrate EHR data to the RDW in the OMOP CDM, and have established Microsoft's Azure cloud environment for its RDW infrastructure with scalability and accessibility. However, the quality of SC RDW is unknown. Thus, how to improve clinician trust so that the data will be accurate and complete or that external tools could be used to analyze them successfully is the major goal for AI Implementation Science research.

We have conducted a comprehensive study using the OHDSI Data Quality Dashboard (DQD), which includes a series of tests to assess SC RDW completeness, plausibility, and conformance to the OMOP CDM so that multi-modality of patient care data quality will be improved and accessible by clinicians and researchers for using AI-Driven clinical decision support. We complemented these tests with an highly-

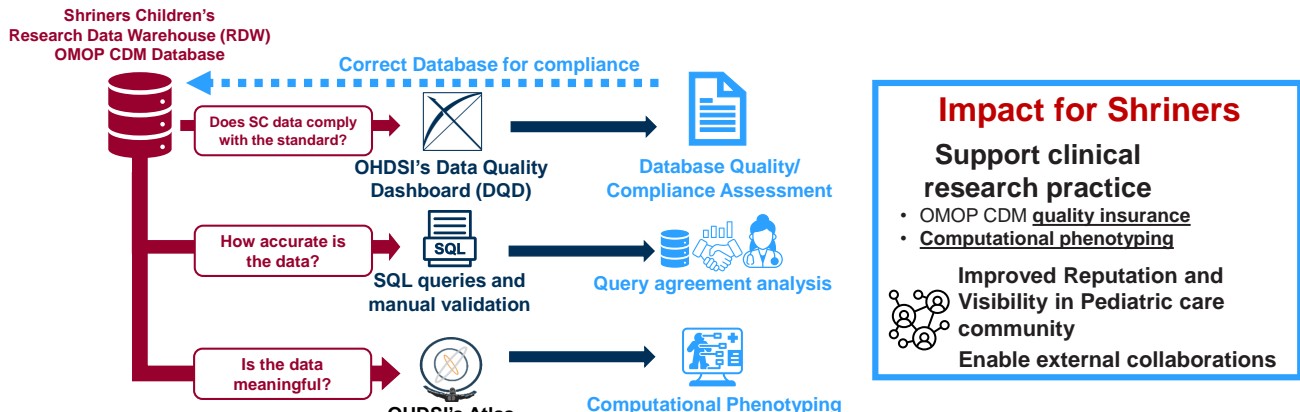

Fig. 1. This is an overview of our current effort towards improving Shriners Children's (SC) Research Data Warehouse (RDW) quality and usability to allow SC's clinicians and researchers to leverage SC's data wealth of unique and rare diseases, to ultimately improve patient care.

interactive dashboard that can be viewed and accessed by researchers as a visual and quantitative feedback mechanism.

The study has three phases as shown in **Figure 1**:

1) Assessing SC's RDW data quality, encompassing database plausibility, completeness, and conformance to the OMOP , and providing visual and quantitative feedback mechanism for researchers.
2) Assessing SC's RDW data query agreement analysis.
3) Assessing SC's RDW usefulness in clinical research by performing computational phenotyping on specific use cases, relying on work done to improve SC's RDW data quality .

## II. PRELIMINARIES

### A. ODHSI's OMOP CDM

The OMOP CDM is an OHDSI-developed health data standard that allows healthcare stakeholders to store and share medical research data [1]. The OMOP CDM uses standard table structures, such as the Person, Condition, and Procedure tables, which store the data point identifier, a concept code specific to the OMOP CDM representing the data, also called Concept ID, and the source data representing the data format before the ETL process. OHDSI developed and used a standard vocabulary to represent the different medical concepts in unique Concept ID codes that are stored in their Concept table[1]. Furthermore, these Concept IDs are categorized by domain, such as the "Condition", "Procedures", "Drugs" domains. This enables researchers to share data in an expected format with expected content, notably to harmonize the various vocabularies and code systems clinicians use to store data.

### B. OHDSI's Data Quality Dashboard

OHDSI built the DQD to allow researchers to evaluate the quality of their OMOP CDM database. OHDSI's DQD is a tool that runs thousands of tests that evaluate the quality

[1]The Concept table, as well as relationship between Concept IDs can be viewed and downloaded from OHDSI's Athena website: https://athena.ohdsi.org/

of a database and compliance with the OMOP CDM [2]. Specifically, DQD is software coded in the R programming language that connects to a SQL database, runs SQL scripts for each quality test, and stores the results in JSON (JavaScript Object Notation) files that can be visualized within a built-in web application that uses R Shiny [3]. OHDSI's DQD database quality tests can be categorized into three metrics categories: Conformance, Completeness, and Plausibility, and into two metric contexts: Verification and Validation [4]. Verification tests are inherently self-contained and do not necessitate the use of an external reference. They examine model constraints, metadata data constraints, system assumptions, and the application of local knowledge. Verification enables us to determine expected values and distributions leveraging intrinsic resources in the local environment. In contrast, validation tests focus on the harmonization of data values in relation to relevant external benchmarks, outside of the local data environment. A potential strategy to establish an external benchmark could involve the amalgamation of results derived from multiple data sites. We summarize the type of test performed for each category in **Table I**.

TABLE I
WE REPRESENT A SUMMARY OF TYPICAL TESTS PERFORMED BY OHDSI'S DQD FOR EACH CATEGORY: CONFORMANCE, COMPLETENESS, PLAUSIBILITY.

| Conformance | Data conforms with format, standards and allowable ranges |
|---|---|
| Completeness | The absence of data is consistent with standards and expectations |
| Plausibility | Data within common and acceptable range (e.g., height and weight >0) |

### C. SC Data Warehouse

The Shriners Health Outcomes Network (SHONet) has established SC RDW system containing data abstraction for clinical efficacy studies and patient cohort count for subsequent clinical research. It includes both previous data from SC Cerner Millennium and newer data from SC Epic System.

| num_violated_rows | pct_violated_rows | num_denominator_rows | check_name | check_level | check_description | cdm_table_name |
|---|---|---|---|---|---|---|
| 751526344 | 0.995999139 | 754545174 | isForeignKey | FIELD | The number and percent of records that have a | OBSERVATION |
| 262206756 | 0.347503059 | 754545174 | isRequired | FIELD | The number and percent of records with a NUL | OBSERVATION |
| 262206756 | 0.347503059 | 754545174 | measureValueCompleteness | FIELD | The number and percent of records with a NUL | OBSERVATION |
| 82657065 | 0.999991023 | 82657807 | cdmDatatype | FIELD | A yes or no value indicating if the MEASUREMEN | MEASUREMENT |
| 27922700 | 0.337810801 | 82657807 | fkDomain | FIELD | The number and percent of records that have a | MEASUREMENT |
| 13821837 | 1 | 13821837 | isRequired | FIELD | The number and percent of records with a NUL | DRUG_EXPOSURE |
| 13821837 | 1 | 13821837 | measureValueCompleteness | FIELD | The number and percent of records with a NUL | DRUG_EXPOSURE |
| 12369550 | 0.894928076 | 13821837 | fkDomain | FIELD | The number and percent of records that have a | DRUG_EXPOSURE |
| 11387683 | 0.137768995 | 82657807 | fkDomain | FIELD | The number and percent of records that have a | MEASUREMENT |
| 8667646 | 0.011487246 | 754545174 | isStandardValidConcept | FIELD | The number and percent of records that do not | OBSERVATION |

**Test Results:**
- **Num_violated_rows:** Number of data points failing the test
- **Pct_violated_rows:** Percentage of data points failing the test
- **Num_denominator_rows:** Total number of data points examined

**Test Descriptors**
- **Type of Test: Validation or Verification**
- **Category of Test: Conformance, Completeness, Plausibility**

Fig. 2. This figure depicts an example of OHDSI's DQD result table after performing tests related to "Source Concept Record Completeness. The results table contains the following information: the number of data points evaluated by the test, the number and percentage of data points that failed the test, and various descriptions of the test.

Currently, new Epic instance are mapped monthly onto the RDW through a standardized ETL process. SC RDW is housed in Microsoft Azure accessible through Azure Virtual Machine via Microsft SQL Management Studio, and primarily used for clinical research. The data are more than 240 Gigabytes (GB) with billions of rows from two sets of tables: one for the data mapped from SC's Cerner and the other for the data mapped from SC's Epic System. SC's RDW access complies with the Health Insurance Portability and Accountability Act (HIPAA), which governs and frames medical data for research, operational, clinical, or educational purposes. Also, SC RDW is controlled via a Role-Based Access Control (RBAC) mechanism, mechanism to ensure data safety and the use of third-party applications and software, streamlining data sharing among clinicians and researchers  Implementation Science

### D. Systematic Challenge in AI Implementation

In real world health system, to incorporate a new third-party software into clinical research, preventing data leak is a top priority. However, it is not clear what may cause data leaking. Thus, we have used engineering simulation strategy to tackle this major challenge. For example, to systematically assess the data quality, we have designed to integrat OHDSI's DQD into SC RDW. To minimize the risk of unintentional data leaks or cyber-attacks, we simulated the SC Azure cloud environment by generating synthetic data that imitating the real-world clinical data. This is one of the widely used approach in AI Implementation Science [5], [6]. This simulated environment was used to evaluate different implementation approaches viability as well as the adaptation required to perform the test on an OMOP CDM version unsupported by OHDSI's DQD.

We used Synthea, a synthetic patient data simulator that we then converted into an OMOP CDM synthetic database

using an ETL process by OHDSI[2] [7]. We ensured that our synthetic OMOP CDM database tables, column names, and column datatypes would match SC RDW.

Now to find systematic quality issues, we have run OHDSI's DQD tests by implementing OHDSI's DQD R code within Microsoft Azure Synapse; implementing the same code within Azure Databricks, and extracting SQL codes from OHDSI's DQD to be run independently (via Microsoft SQL Server Management Studio (SSMS)).

*1) AI Implementation Barrier: Lack of Compatibility within Constained Environment:* WE could not run OHDSI's DQD R code due to the impossibility of installing a critical package: "rJava" in Azure Synapse and Databricks. Even though Microsoft Azure Synapse and Azure Databricks allows the creation of a secure environment, where developers can implement and run codes while letting Microsoft handle data access security, however, if the environment does not allow the installation, it becomes nonfuncational. This prompted us to realize one AI Implementation issue: **constraints of environment**. Because OHDSI's DQD allows the user to extract the individual SQL scripts to run the tests directly on the database, we directly running SQL scripts on SC RDW.

### E. Systematic Evaluation Metric in AI Implementation

While implementation AI infrastructure, it is critical to find systematic issue that can have major impact in the AI system. In SC RDW case study, one major issue in data harmonization is the quality. The systematic investigation has systematic evaluation metric as captured in **Figure 2**:

- The number of data points evaluated by the test.
- The number and percentage of data points that failed the tests.
- Data regarding the script running time

[2]The code developed by OHDSI can be found using the following link: https://github.com/OHDSI/ETL-Synthea

TABLE II

THIS FIGURE DEPICTS THE RESULTS OF OHDSI'S DQD TESTS ON SC'S RDW DATA MAPPED FROM ITS CERNER EHR.

| Category | Verification | | | | | Validation | | | | | Total | | | | |
|---|---|---|---|---|---|---|---|---|---|---|---|---|---|---|---|
| Type | Pass | Fail | Removed | Total | % Pass | Pass | Fail | Removed | Total | % Pass | Pass | Fail | Removed | Total | % Pass |
| Plausibility | 78 | *3* | *256* | 337 | 96.3% | 275 | *12* | *0* | 287 | 95.8% | 353 | *15* | *256* | 624 | 95.9% |
| Conformance | 437 | *159* | *43* | 639 | 73.3% | 62 | *20* | *17* | 99 | 75.6% | 499 | *179* | *60* | 738 | 73.6% |
| Completeness | 241 | *23* | *84* | 348 | 91.3% | 10 | *5* | *1* | 16 | 66.7% | 251 | *28* | *85* | 364 | 90.0% |
| **Total** | 756 | *185* | *383* | 1324 | 80.3% | 347 | *37* | *18* | 402 | 90.4% | 1103 | *222* | *401* | 1726 | **83.2%** |

- Various descriptions regarding the test, such as the check level, which table was evalutated by the test, a human-readable description of the test itself, and the category and context of the test

We extracted the SQL codes used to perform the quality tests using an offline and local implementation of OHDSI's DQD R code (via a specific built-in function). OHDSI's DQD only supports OMOP CDM v5.2 and onward; therefore, we had to adapt the SQL code to match SC RDW OMOP CDM v5.1. This adaptation task consisted in three steps:

1) We first clearly identified the differences between SC's OMOP CDM implementation (using our simulated environment) and the OMOP CDM v5.2
2) We then removed tests performed on tables and columns not present in SC's OMOP CDM. In the case a table or column in SC's OMOP CDM closely matched a counterpart in OMOP CDM v5.2, we adapted the test to be performed on the affected table or column. However, to maintain a fair comparison with other research results, we failed the specific test(s) corresponding to checking whether the table or column is present within the database. Our goal was to perform as many tests as possible, hence our modifications.
3) As Microsoft SQL Server Management Studio is sensitive to table and column capitalization, we capitalize the names in all SQL scripts. We noticed that the OMOP CDM is not standardize letter capitalization.

We conducted 1,125 tests, with each test covering from a few dozen to about 750 million data points. The test results in **Table II** show a pass rate of **83.2%**, with a total of 222 failed tests and 401 removed tests. Most of these failed tests were for conformance verification (159 failed tests and 43 removed tests), completeness verification (23 failed and 84 removed), conformance validation (20 failed and 17 removed), plausibility validation (12 failed and 0 removed), completeness verification (5 failed, 1 removed) and plausibility verification (3 failed and 256 removed). Because there are missing tables and columns in SC RDW, we had to manually remove 222 tests from the SQL scripts and categorized them as "test not run".

*F. Comparison with similar research*

We compared in **Figure III** SC's RDW quality with an European consortium comprising 22 partners, The European Health Data & Evidence Network (EHDEN), who used OHDSI's DQD to assess the quality of each partner's OMOP

CDM database. We noticed that SC results are similar to EHDEN's results, with a similar number of tests performed. As expected, SC' RDW OMOP CDM failed numerous tests related to the "structure" of the database (columns, tables, datatypes).

*G. Quality Evaluation Results Analysis in AI Implementation Science*

We analyzed and explored the potential source of failed and removed tests and extracted five sources:

- Different OMOP CDM version used (different tables, columns)
- Missing Values, either null or replaced with placeholder value
- Concepts code not conformant with the OMOP CDM
- Unexpected concept code for certain columns
- Unexpected data for certain procedure/ condition compared with the visit date

*1) Removed Tests:* As mentioned above, we had to remove 401 tests that were causing errors when running Microsoft SQL Management Studio. We identified two major causes of errors:

1) Missing tables and columns as Shriners' RDW follows an earlier version of the OMOP CDM
2) Missing or "*null*" values in certain columns, which would throw an error for columns with datatypes "time" or "datetime". This would also happen with specific columns for unknown reasons, such as the "*condition_status_source_value*" column in the condition table.

We have detailed the number of removed test for each SQL script run in **Table IV**.

*2) Verification:*

*a) Plausibility:* We observed three plausibility verification tests that failed, although less than 4% of data points failed the tests. The first failed tests concerned data points with less than one procedure ordered (null or negative). The other failed tests were for the condition and the visit with an end date after the patient's death.

*b) Conformance:* The most prevalent type of error was the nonexistance of expected tables and columns in the OMOP CDM v5.2. This type of error is mostly impacting data interoperability, and can show a lack of data for AI Implementation.Another source of error was the nonconformance of a certain Concept ID domain. For example, we observed that 100% of the Death Type Concept IDs were not in the "type" domain, 99% of the gender Concept ID in the provider

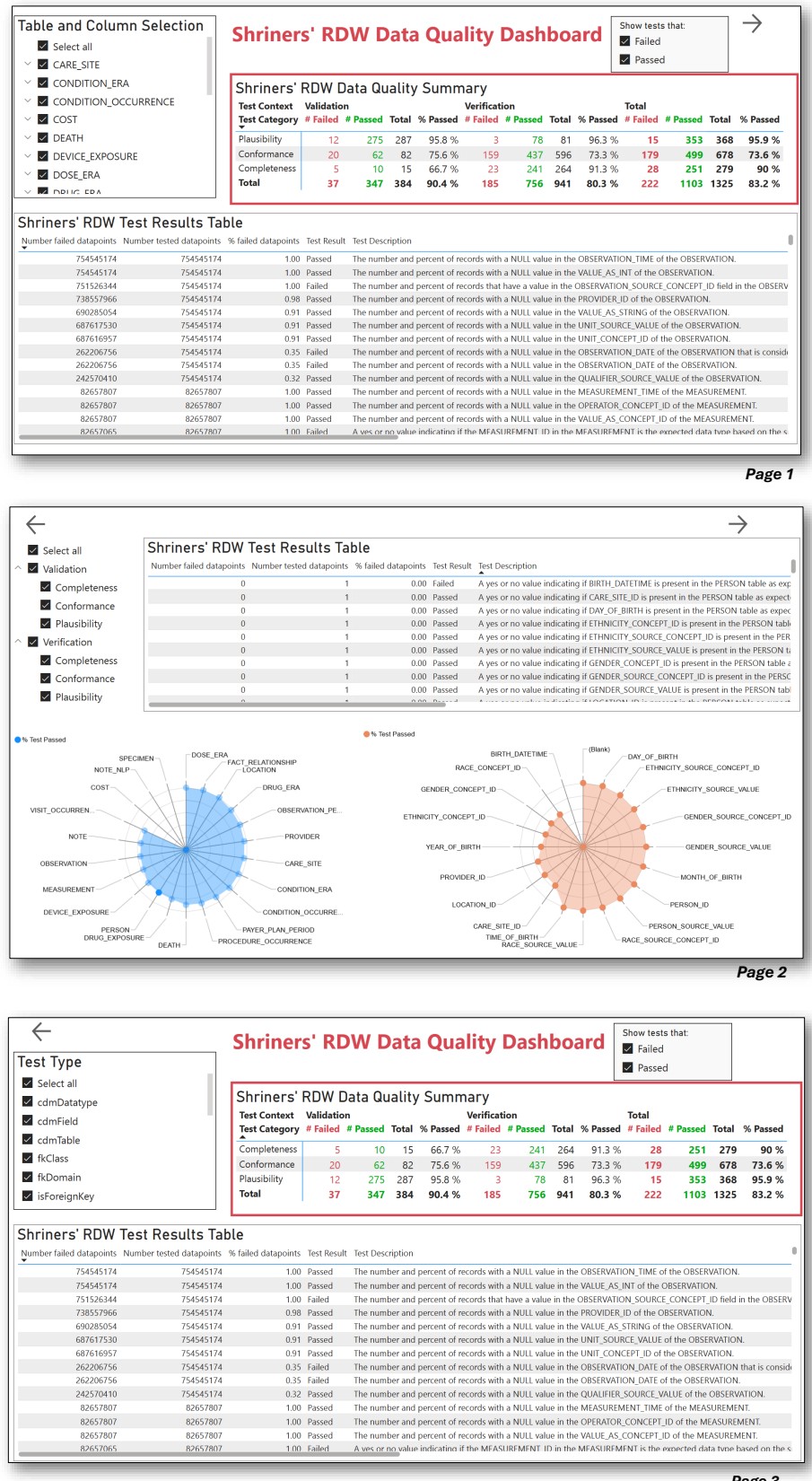

Fig. 3. Dashboard created within Microsoft Fabric and with Power BI to represent the results from the DQD tests. The design is inspired from OHDSI's original DQD dashboard, but with additional interactivity and visualization.

TABLE III
COMPARISON OF THE TESTS RESULTS BETWEEN SHRINERS' OMOP CDM AND EHDEN FIRST DQD RUN. WE REPRESENTED IN RED THE TESTS FOR WHICH SHRINERS' RESULTS ARE AT LEAST 10% UNDER EHDEN SCORES, AND IN GREEN TESTS FOR WHICH SHRINERS RESULTS ARE AT LEAST 10% ABOVE EHDEN SCORES.

| Test Type | Nb Test (*ours*) | % Passed (*ours*) | Nb Test Total (*EHDEN*) | Nb Test per Site (Mean) (*EHDEN*) | % Passed (*EHDEN*) |
|---|---|---|---|---|---|
| cdmDatatype | 129 | 80.6% | 1147 | 76 | 99.7% |
| cdmField | 270 | 70.4% | 4341 | 289 | 97.1% |
| cdmTable | 22 | 86.4% | N/A | N/A | N/A |
| fkClass | 2 | **100%** | 12 | 0 | 66.7% |
| fkDomain | 27 | 48.1% | 332 | 22 | 70.8% |
| isForeignKey | 88 | 73.9% | 1116 | 74 | 77.2% |
| isPrimaryKey | 16 | 93.8% | 162 | 10 | 98.1% |
| isRequired | 82 | 75.6% | 768 | 51 | 96.4% |
| isStandardValidConcept | 35 | 71.4% | 407 | 27 | 84.5% |
| measureConditionEraCompleteness | 1 | 0% | N/A | N/A | N/A |
| measurePersonCompleteness | 14 | 71.4% | 216 | 14 | 81.5% |
| measureValueCompleteness | 200 | 90% | 2313 | 154 | 97% |
| plausibleDuringLife | 14 | **85.7%** | 207 | 13 | 70.5% |
| plausibleGender | 287 | **95.8%** | 1228 | 81 | 82.8% |
| plausibleTemporalAfter | 7 | **100%** | 334 | 22 | 79.3% |
| plausibleValueHigh | 27 | 100% | 406 | 27 | 91.6% |
| plausibleValueLow | 33 | 97% | 435 | 29 | 88% |
| sourceConceptRecordCompleteness | 12 | 91.7% | 141 | 9 | 90.8% |
| sourceValueCompleteness | 18 | 100% | 254 | 16 | 93.3% |
| standardConceptRecordCompleteness | 34 | 94.1% | 381 | 25 | 85.8% |
| withinVisitDates | 7 | 57.1% | N/A | N/A | N/A |
| **Total** | **1325** | **83.2%** | **14200** | **939** | **92.8% ± 6.3%** |

TABLE IV
TESTS REMOVED DUE TO MISSING TABLES AND COLUMNS IN SHRINERS' OMOP CDM DATABASE. WE HIGHLIGHTED IN BLUE THE VALIDATION TESTS AND IN GREEN THE VERIFICATION TESTS. THE SQL SCRIPTS ARE SORTED BY NUMBER OF TESTS REMOVED.

| Scripts Name | # Removed Tests | % Removed Tests | # Total Tests | Category |
|---|---|---|---|---|
| CONCEPT_plausibleUnitConceptIds | 181 | 100% | 181 | Plausibility |
| FIELD_measureValueCompleteness | 68 | 25% | 268 | Completeness |
| FIELD_plausibleTemporalAfter | 29 | 81% | 36 | Plausibility |
| FIELD_isForeignKey | 23 | 21% | 111 | Conformance |
| FIELD_isRequired | 17 | 17% | 99 | Conformance |
| FIELD_plausibleValueLow | 18 | 35% | 51 | Plausibility |
| FIELD_plausibleValueHigh | 18 | 40% | 45 | Plausibility |
| FIELD_isStandardValidConcept | 13 | 27% | 48 | Conformance |
| FIELD_plausibleDuringLife | 10 | 42% | 24 | Plausibility |
| FIELD_sourceValueCompleteness | 9 | 33% | 27 | Completeness |
| FIELD_standardConceptRecordCompleteness | 7 | 17% | 41 | Completeness |
| FIELD_isPrimaryKey | 4 | 20% | 20 | Conformance |
| FIELD_fkDomain | 3 | 10% | 30 | Conformance |
| TABLE_measurePersonCompleteness | 1 | 7% | 15 | Completeness |
| CONCEPT_plausibleGender | 0 | 0% | 287 | Plausibility |
| FIELD_cdmField | 0 | 0% | 270 | Conformance |
| FIELD_cdmDatatype | 0 | 0% | 129 | Conformance |
| TABLE_cdmTable | 0 | 0% | 22 | Conformance |
| FIELD_sourceConceptRecordCompleteness | 0 | 0% | 12 | Completeness |
| FIELD_withinVisitDates | 0 | 0% | 7 | Conformance |
| FIELD_fkClass | 0 | 0% | 2 | Conformance |
| TABLE_measureConditionEraCompleteness | 0 | 0% | 1 | Completeness |

table were not in the "gender" domain, or 90% of the route Concept ID in the drug exposure table were not in the "route" domain. Another type of failed test related to certain values with the wrong data type, such as 99% of measurement IDs. Lastly, some records were not recorded within a week of their corresponding visits, which is the case for 17% of notes. These errors can lead to poor data quality for AI Implementation, which might rely on untrustworthy data to make its prediction.

*c) Completeness:* Completeness verification tests evaluated the number of "*null*" data for certain tables' columns. This was the case for 100% of drug type and "device type Concept IDs" in the drug and device exposure tables, respectively. This

was also the case for 38% of the device exposure start date in the device exposure table, 35% of the visit Concept ID in the visit occurrence table, and the observation date in the observation table. A total of 23 columns failed this series of tests. These errors are linked with sparsity in the data, which adds challenge in AI Implementation, since data will have to be interpolated in some instance.

*3) Validation:*

*a) Plausibility:* Only a dozen data points failed plausibility validation tests. Specifically, this series of tests checks whether certain patients' conditions are plausible, mostly related to their gender. For example, a missing or unexpected gender in the data could lead to a failed data point. However, the number of data points affected by these tests ranged from one to 36 patients, with only one to six data points failing.

*b) Conformance:* We noticed that the failed completeness verification tests and the conformance validation tests were very similar: the former checks for missing values, and the latter checks for missing values that should not be "*null*".

*c) Completeness:* Completeness validation tests checked the percentage of patients with data in other tables. For instance, 100% of the patients do not have data in the observation period, drug era, and condition era tables. Also 98% of patients do not have data in the drug exposure table. We show the most prevalent failed test in **Figure 4**.

## III. POWER BI DASHBOARD

We build a prototype Power BI dashboard using the results of the tests. Our goal was to obtain an overview summary of the results based on OHDSI's DQD web application, that could be easily implemented within a secure cloud-based healthcare environemnt. Moreover, our dashboard has additional functionalities and options to visualize the results for specific tables

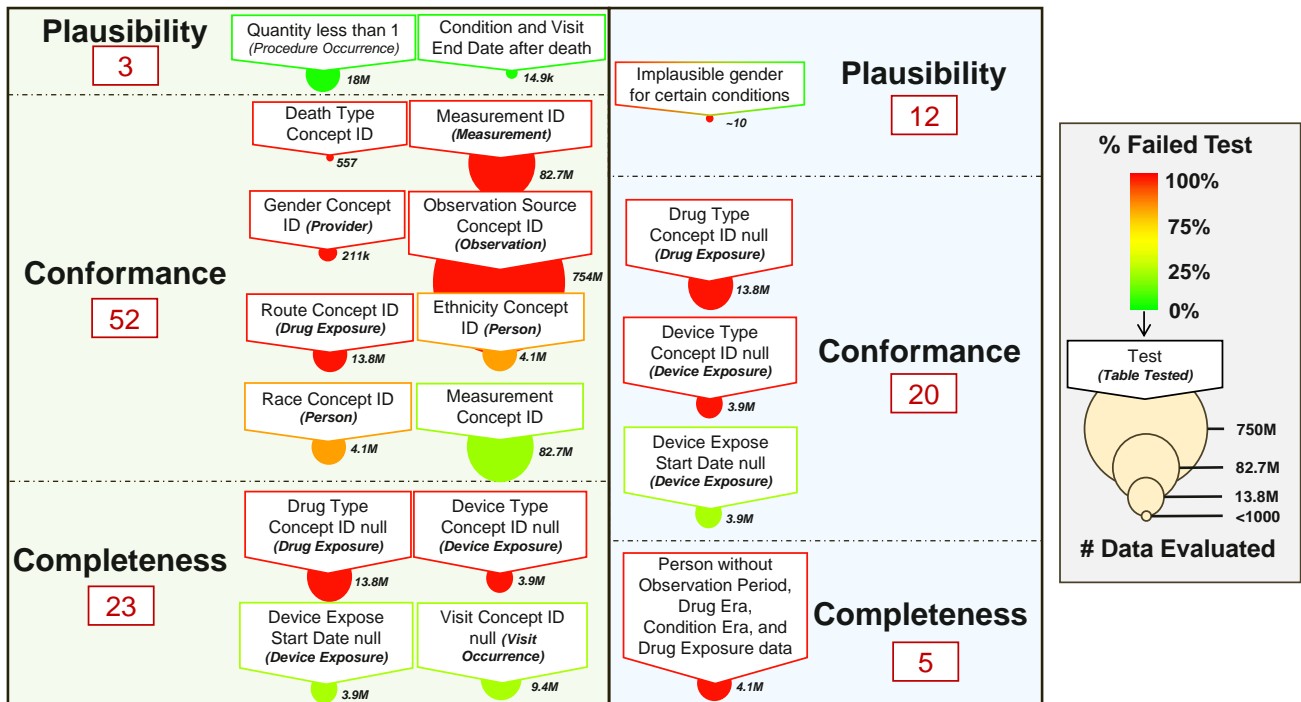

Fig. 4. This figure depicts the most prevalent failed tests when running OHDSI's DQD on SC's OMOP CDM database mapped from Cerner. The color represents the percentage of data points failing the test, and the size of the circle represents the number of data points that were evaluated.

and columns, test categories and contexts, or test types to enable SC's researchers to evaluate the data quality at a higher granular level. We show the dashboard in **Figure 3**.

## IV. DISUSSIONS OF CASE STUDY

We focus on AI Implementation Science research by using SC RDW as one case study. The quality in any AI for medicine is critical. In this case study, we found that the SC RDW quality and compliance with OMOP CDM is consistent with the literature. For example, EHDEN, evaluated the implementation and improvement capability of OHDSI's DQD in 15 different hospitals, and used OHDSI's DQD to improve their healthcare network through an iterative process [8], [9]. They successfully identified failed tests and their corresponding data points and then used the insights gained through this process to improve the ETL process, subsequently improving the quality and compliance of their OMOP CDM database. Similarly, Ward et al. used OHDSI's DQD to assess the quality of their ongoing effort toward building a medical research database in Australia [10]. Peng et al. used OHDSI's DQD for a similar goal in Germany [11]. More recently, Bhattacharjee et al. utilized the full suite of OHDSI's tools to build an OMOP CDM database for an African Population part of the Implementation Network for Sharing Population Information from Research Entities (INSPIRE), specifically OHDSI's DQD to assess the quality of their newly built database. In each instance, OHDSI's DQD was ultimately used to improve database quality and compliance with the OMOP CDM, notably via the improvement of the ETL process.

SC RDW's source of failure may be attributed to multiple points in the dataflow pipeline, not only in the ETL process. Therefore, we not only aim at using OHDSI's DQD to improve SC OMOP CDM database but also aim to improve the source database. For example, we might be able to detect procedures, notes, or condition dates that are impossible.From AI Implementation Science perspective, systematic finding of problems during implementation is critical. Also finding solutions to improve data quality is essential for AI to have true impact in healthcare. s [11], [12].

Most of the failures reported by OHDSI's DQD were expected, specifically when tested on tables and columns with a substantial proportion of missing data. Moreover, most of these errors were inconsequential for current research within SC as SC's RDW data is usually manually queried by SC's data engineers. Hence the scope of this work is to enable SC to directly use tools developed by or for an OMOP CDM database (such as OHDSI's Altas) as well as to understand the mechanism we could use to obtain quantitative and visual feedback related to the data quality.

### A. Future Work

Future projects aiming at implementing OHDSI's DQD within a real-world healthcare system should take into account the following:

1) There are no standards regarding tables and columns name letter case, meaning that databases sensitive to letter case might throw errors when running the tests
2) Some of the tests checking the conformance of certain tables and columns' names and existence would not run properly in the case of a missing table or column. In some cases, this would make the test assessing the existence of tables and columns obsolete. Indeed, these tests would either run and be validated or throw an error.

We also aim to improve SC database quality in two steps:

1) Streamlined retrieval of data points failing the tests
2) Identification of failure point (is the failure due to the Extract, Load, Transfer process, earlier or later in the pipeline.

Ultimately, we recommend any developer building an application based on the OMOP CDM to expect an imperfect database and implement robustness to errors and missing data in their application. Future efforts will focus on identifying data points for each failed test, as well as points of failure in the data pipeline. All the results we obtained were from the data mapped from previous SC's Cerner Millenium EHR, we will also assess the quality of the database mapped from Epic System. Moreover, our goal is to assess the quality of data queries and the usefulness of data once we have finished our current efforts.

## V. Conclusions

In this case study, we conducted comprehensive study of AI Implementation Science by working on a 22-hospital international children's health system. We implemented OHDSI's DQD, and assessed Shriners' RDW database quality. We successfully used OHDSI's DQD to identify the sources of non-compliance of a real-world healthcare OMOP CDM database. We reported the completeness and conformance of the data to the Observational Medical Outcomes Partnership (OMOP) Common Data Model (CDM), and the data content plausibility. We identified the patterns of failure and the underlying causes. With these major implementation science issues discovered and solved, we not only improved the access and quality of SC RDW data to enable clinicians and researchers to more trustworthy decision support for better patient care outcome, but more importantly, we have developed an entire workflow and pipeline that can be extended and utilized for other healthcare system.

## Conflict of Interest Statement

The authors declare that the research was conducted in the absence of any commercial or financial relationships that could be construed as a potential conflict of interest.

## Internal Review Board Note

This project was undertaken as a Quality Improvement Initiative at Shriners Hospitals for Children and, as such, was not formally supervised by an Institutional Review Board.

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
