# OpenReview forum: "Improving A Large Healthcare System Research Data Warehouse Using OHDSI’s Data Quality Dashboard"
_IEEE.org/EMBS/BHI/2024/Conference — IEEE BHI'24_

### Official Review · Reviewer_Vn5z · 2024-08-08
**An initial step in evaluation of a hospital system’s raw research data, demonstrating what may be common pitfalls and difficulties in correcting errors and ensuring standardization of data.**

**Overall Rating:** 7
**Confidence:** 3

**Other Quality Metrics:**

a.	Clarity of writing: Great

b.	Clinical Significance: Fair

c.	Methodological Novelty: Fair

d.	Experiments and Results: Fair

**Questions For The Authors:**

The paper states the many of the errors seen in the DQD are inconsequential for the current research. Are these errors more due to formatting and thereby causing incompatibility with the standard which could be fixed?

**Strengths:**

The paper is very straightforward and easy to understand. For the errors identified in this initial phase, the paper proposes reasonable improvements that can be made to improve standardization and ultimately usefulness of the Shriners’ RDW. These are also likely sources of errors and areas of improvement for other institutions looking to perform a similar task.

**Summary Of The Paper:**

The paper utilizes the OHDSI DQD tool to assess Shriners’ collection of raw data after migration to an electronic health record system and using OHDSI’s Observational medical Outcomes Partnership Common Data Model standard. The paper tests this Raw Data Warehouse (RDW) for plausibility, completeness, conformance to this standard. This is the first step in a three-phase project intended to test compliance to the standard, then accuracy of the data, and finally the ability to use the data for meaningful research. Most of the resulting errors came from missing or obsolete tables and columns, missing values, non-conforming concept codes, and unexpected concept code and data. The paper removed tests that resulted from missing tables and columns due to the RDW following an earlier version of the standard and unknown errors tied to time datatypes. Completeness validation showed that 100% of patients lacked data in observation period, drug era, and condition era tables. 98% did not have drug exposure data.

**Weaknesses:**

This paper discusses an initial first step in evaluating a dataset.

I may be misunderstanding, but the reason and potential influence of the removed tests may need to be further investigated or explained.

In section III. Original Work, C. Test Results Analysis, first bullet point: should Tables, Columns, and Values be capitalized?

---

### Official Review · Reviewer_DyjZ · 2024-08-17

**Overall Rating:** 6
**Confidence:** 3

**Other Quality Metrics:**

(a) Clarity of Writing: good. I understand that the paper uses a lot of abbreviations and necessarily so, but I had to go back and re-read what these abbreviations meant.

(b) Clinical Significance: fair. It had synthetically generated data. So it’s reproducibility in real-world clinical applications seems hard to judge.

(c) Methodological novelty: good. Implementing of the dashboard is a novel solution and the areas of testing are a useful contribution.

(d) Experiments and Results: fair. I think there are potential actionable insights, but they could be better compiled into a set of best practices.

**Questions For The Authors:**

How exactly are you ensuring that the synthetic data generated is closely matching how real-world data has missing data?

In other words, what assumptions are used about the randomness of the missing data?

**Strengths:**

1. The study offers clear evaluation criteria to assess the quality of the RDW.

2. The study also states clear sources of data issues such as missing tables, obsolete columns, and non–conformant concept codes.

**Summary Of The Paper:**

This paper explores the enhancement of Shriners Childern’s (SC) Research Data Warehouse (RDW) quality and usability through the implementation of the Observational Health Data Sciences and Informatics (OHDSI) Data Quality Dashboard (DQD). The main contribution of this work is to address issues like data incompleteness, non-conformance to standards, and usability limitations. The authors used OHDSI’s DQD to assess the RDW’s data quality across different areas like completeness, conformance, and plausibility. They uncover areas that need improvement such as missing tables, obsolete columns, and non-conformant concept code

**Weaknesses:**

1. I believe that this paper covers very specific problems related to SC’s RDW. This paper reads more like a technical report than a research paper. I am uncertain if this work is very reproducible outside SC’s RDW infrastructure.

2. Moreover, the study relies on synthetic data to test the implementation of OHDSI’s DQD due to security concerns. This adds another layer of separation from real-world applicability.

3. This paper mentions different methods for implementing the DQD, but it does not provide a comparative analysis or benchmark. Some of the methods were chosen simply because critical packages weren’t able to be installed. It also lacks explanations on how to use the DQD. A visualization of the dashboard would be helpful.

4. This paper should spend more time discussing best practices for Data Warehouse managers in mitigating such issues.

5. The broader implications of this work for healthcare researchers who are using such Data Warehouses should also be mentioned in more detail. How should they better query the data?

---

### Official Review · Reviewer_Zvy5 · 2024-08-18
**An overview of data quality checks used for Shriners' Children's' research data**

**Overall Rating:** 6
**Confidence:** 4

**Other Quality Metrics:**

(a) Clarity of writing: Good
(b) Clinical Significance: fair
(c) Methodological Novelty: fair
(d) Experiments and Results: good

**Questions For The Authors:**

In the "II. Background" section, it says that one of the things verification tests examine is "application of local knowledge", but it was not clear to me what this referred to. Is there another way to say this that a reader who's unfamiliar with OHDSI can understand?

**Strengths:**

The paper offered an overview of common data issues that researchers encounter in real world databases. It offers one tool to analyze the data quality for medical systems.

**Summary Of The Paper:**

The paper describes the authors process of applying the OHDSI data quality approach on Shriners' Children's (SC) data warehouse. The authors motivation to run these checks was to uncover any data that do not follow these 3 standards: conformance, completeness, and plausibility.

The process to enable the OHDSI data quality dashboard (DQD) on SC's environment involved several workarounds due to compliance and regulation constraints. The main workarounds were recreating a similar locked-down environment for development and testing and rewriting the OHDSI DQD R code to SQL, since SC's environment did not allow the use of rJava.

The SQL scripts were then run on the actual data warehouse, and the findings were summarized in this paper. A high number of "failed tests" were found in this process, indicating malformed or missing data.

**Weaknesses:**

A concern with rewriting code from one language to another is testing if the code behaves the same way. I think an overview of how the authors validated the parity of the new SQL scripts with the R code would be useful to have more trust in the results.

One thing that could be improved upon is more explanation on why it's important to do this data quality evaluation.

The paper could have also chosen to list some remediation steps to take that could improve the data quality, but perhaps that was not in scope for this paper.

---

### Decision · Program_Chairs · 2024-09-23

Accept